# Textbooks and Learning Materials in Physical Education in the International Context: Literature Review

**DOI:** 10.3390/ijerph19127206

**Published:** 2022-06-12

**Authors:** Jesús Rodríguez Rodríguez, Denébola Álvarez-Seoane, Víctor Arufe-Giráldez, Rubén Navarro-Patón, Alberto Sanmiguel-Rodríguez

**Affiliations:** 1Facultad de Ciencias de la Educación, University of Santiago de Compostela, 15782 Santiago de Compostela, Spain; jesus.rodriguez.rodriguez@usc.es; 2Grupo de Investigación Stellae, 15782 Santiago de Compostela, Spain; carmendenebola@gmail.com; 3Facultad de Ciencias de la Educación, Campus de Elviña, University of A Coruña, 15071 A Coruña, Spain; v.arufe@udc.es; 4Facultad de Formación del Profesorado, University of Santiago de Compostela, 27001 Lugo, Spain; ruben.navarro.paton@usc.es; 5Faculty of Language and Education, University of Camilo José Cela, 28692 Madrid, Spain

**Keywords:** curricular resources, didactic materials, digital resources, educational resource, physical education, textbooks

## Abstract

This paper presents a review of the research carried out internationally in recent years regarding textbooks, teaching materials, and the field of physical education. The objectives to which this article aims are as follows: To offer a compilation of current research papers published in the Scopus, Web of Science, and Dialnet databases and to provide a classification of the research lines undertaken on the subject. The information used is based on a review of studies listed in Scopus, Web of Science, and Dialnet. For this, a systematic review was carried out with the terms: “curricular resources”, “didactic materials”, “digital resources”, “educational resource”, “physical education”, and “textbooks”, allowing the selection of original articles, which included information on this line of research. For the literature review, only articles that presented scientific evidence were used, discarding merely descriptive articles or didactic experiences that were not investigated with the scientific protocol. A total of 589 articles were found, although after applying the review’s inclusion criteria, they were reduced to 54 articles. The investigations do not follow a common protocol and the researchers address multiple objectives in them, highlighting the studies on the implicit ideological discourse in the images of physical education textbooks. The results reveal a diversity of research, in particular, studies involving ideological discourse. Insofar as methodology, content analysis of textbooks and materials was the most common approach.

## 1. Introduction

In recent decades there has been a considerable advance in the general study of textbooks and didactic materials in the international context from a variety of perspectives [1,2]. In particular and within the scope of the current analysis, a good deal of research regarding the study of these resources in the field of Physical Education (PE) has proliferated [3]. We should also point out the proliferation of proposals and didactic resources elaborated by teachers themselves with a variety of characteristics and formats, whether manipulative, printed, digital or hybrid [4]. The following consideration by García and Ruíz [5] clarifies the concept of material. The materials, as physical elements that they are, have a considerable influence on the teaching-learning process and can greatly condition its effects. Among these are: the physical space in which the sessions take place, the social environment; the attention given to the subject within the school; previous experiences of the students; familiarity of students and teachers with the materials; age level and learning ability; cultural tradition, transmission of gender stereotypes and teachers’ own conceptions about how to understand and work with curricular materials [5].

–Regarding the role that didactic materials can play, the following suggestions by Díaz [6] can help to contextualize the scope under consideration:–Motivational function: materials should be able to maintain students’ attention through the allure of shapes, colour, touch, actions, sensations, and so on.–Structuring function: materials should constitute a medium between reality and knowledge, insofar as they fulfill organizational functions and even constitute an alternative to reality itself.–Strictly didactic function: it is essential for there to be coherence between the material resources to be used and the objectives and contents to be taught.–Facilitating learning: in PE many types of learning would not be possible without the existence of certain resources and materials, some of which constitute an essential learning facilitator.–Teacher support function: teachers need to use resources that facilitate the teaching task in aspects such as planning, teaching, assessment, data recording, and control.

Among the diversity of resources in the field, textbooks stand out for their role and influence in the development of the PE curriculum. This teaching medium has previously been analysed [7] and is considered to be an artefact with strong political and ideological connotations. It has been the object of analyses and prospective studies for decades. How it is conceptualized and the way of understanding its uses and implications for teaching represents a way of defining the school and the role of the teacher. Many consider it to be the only source of concretion for the curriculum and some of us believe that it should be seriously re-conceived to give way to a more innovative type of educational approach focusing on teachers’ professional development and attention to the student diversity (p. 10). With specific reference to PE, various authors have pointed out the need for studying the particularities of this area [8].

Here we carry out a bibliographic review and classification of existing research lines in order to provide researchers with an overview of available studies and a summary of their main conclusions as well as insights into future trends and research needs. In the case of PE, the fact is that only a scant number of reviews have been carried and we have only found the odd classification in a specific country [3].

The classification proposals and reflections by Pere, Devís, and Peiró [9] and more recently by Molina [10] have been important references upon which to structure our work. To these definition and classification proposals must be added the increasingly prevalent digital materials and online resource proposals that represent a different way of understanding teaching reality [7,11,12,13]. In particular, we observe that the number of papers related to the study of didactic and curricular materials has proliferated in recent years. We should also highlight the proliferation of didactic proposals and resources prepared by teaching staff, which present diverse characteristics and a variety of formats, whether manipulative, printed, digital or hybrid.

Ours is a broad concept of didactic material and educational resources that takes into account the value of diversity and understands the importance of textbooks (Rodriguez and Rodriguez, 2016). In the specific case of PE, various authors have highlighted the need to study and analyse the particularities of textbooks and other teaching media in this field [3,8].

The objectives to which this article aims are as follows:(a)To offer a compilation of current research papers published in the Scopus, Web of Science, and Dialnet databases on teaching materials in the area of PE in the international arena.(b)To provide a classification of the research lines undertaken on the subject.

## 2. Materials and Methods

The documentary search and study selection were guided by four premises. Firstly, under the pretext of carrying out an international review, we looked for papers published in high-impact journals indexed in Scopus, Web of Science (WoS), and Dialnet. Secondly, several search descriptors were established according to the usual nomenclature and recommendations from the European Thesaurus of Education [*didactic material, curricular material, digital didactic material, textbook, educational resource, educational technology*], which were crossed with the term “*physical education*”. Thirdly, the search and selection were limited to papers published in the period 2000–2021 to focus on the most recent developments in the field; exceptionally, previous papers were included when they served to contrast or corroborate more recent reports. We have referred to articles published in journals indexed in the databases cited above. Though exceptionally, a doctoral thesis that stood out for its singularity and representation in relation to the aforementioned field was also considered (Table 1).

The information search consisted of four differentiated phases (Figure 1). The first stage involved searching for articles based on descriptors from the European Thesaurus of Education. The second stage involved searching the databases using the previously described inclusion criteria. The third stage was the analysis of articles and classification into themes; and finally, the fourth stage was the elaboration of a systematic review. The set of papers and studies reviewed using content analysis have been classified into seven categories looking for links between the different objects of study and serving to configure the research lines (the results are presented in these categories). It should be noted that there are numerous publications on Information and Communication Technology (ICT) and PE that have not been included in this article because their diversity and length require specific treatment: Nevertheless, digital resources are presented in the references of the literature in the identified categories. Despite being a literature review, an attempt has been made to meet the maximum number of criteria recommended by the PRISMA protocol for systematic review and meta-analysis articles [14].

After a preliminary review of 70 documents, articles that did not directly address or were not associated with textbooks and learning materials in PE were discarded. Ten documents are removed for other reasons, e.g., scientific documents that deal with the PE but without being contextualised in textbooks or teaching materials. Following the application of the selection criteria and categorisation, a total of 54 articles remained which provided details of the scientific method followed.

## 3. Results

After the review of textbooks and learning materials in PE was conducted as shown in the flowchart, the result was 54 publications. All of them underwent a categorization process by subject, yielding five analysis categories. The publications resulting from the review have been classified into five categories (Table 2).

## 4. Discussion

The following is a summary of the main lines of research reviewed; the varying degree of development of each line can be deduced from the analysis.


*(a) Studies on the use and characteristics of didactic materials and textbooks in PE*


The studies on the use of didactic materials in the area of PE or the teaching-learning processes they promote present a range of concerns. Two basic types of studies have been identified in this area: (a) articles that focus on the motivational aspect of digital resources, and (b), articles that analyse which resources are most widely present and what they are used for.

Regarding articles that focus on the motivational aspect of digital resources, in a study on material formats and the most widely used teaching media together with their proper use to promote comprehensive student development, Perazas, Gil, Pardo, and Soler [15] initially emphasize that teaching media include all those components of the educational process that act as material aids for the instructional or educational methods used to achieve learning objectives. Nevertheless, these media cannot replace teachers’ educational and human function, because teachers are who direct, organize and control the educational process. For these authors [15], teaching aids help to promote student confidence, contribute to the emotional aspect of knowledge and reinforce collective work. Therefore, the authors maintain that didactic materials are an important component of the teaching-learning process in PE and proper use guarantees the quality of the process. These authors also emphasize that digital learning materials provide students with greater motivation, humanize teachers’ work and favour the transmission of knowledge with a more scientific approach.

In the specific case of teaching fencing in schools, teachers point out that materials correctly guide the process insofar as the functions of structuring, motivating, supporting, and facilitating learning [16]. In addition, 86.8% of teachers believed that the lack of teaching materials for fencing was the problem with respect to choosing this sport as content for PE.

Regarding the analysis of the most widely present resources and what they are used for, one of the most notable studies by Peiró-Velert, Molina-Alventosa, Kirk, and Devís-Devís [17] involved the use of printed curricular materials by PE teachers in 310 secondary schools in Spain. These authors reported that printed teacher support resources in PE such as data logging materials were very often used for tasks such as recording attendance and that students’ textbooks were used for observation notes. Classroom journals were the most widely used materials by teachers with less experience, while textbooks were the most widely used materials by more experienced teachers [17].

In the same line of research, Barroso and Darido [18] analysed the modifications made by PE teachers to the didactic material for teaching the sport classification system in eighth and ninth grades, and whether these changes would suit other teachers working in the same education level. The method used was action research involving three categories: understanding and evaluating the sport classification system, understanding and evaluating the didactic material, and implementing the didactic material on the sports classification system. The first demonstrated that the positive factors were greater than the possible difficulties, the second indicated that teachers were favorable to the use of pedagogical books and the third revealed the importance of materials being adaptable. These authors [18] also observed that teachers were in favor of using didactic books during pedagogical practice, as long as the teacher is free to substitute exercises, activities, tasks, and other factors in order to provide pedagogical flexibility. They maintained that it was essential for teachers to understand the need for making modifications to suit their educational reality on the premise that any material is improved and made more appropriate when teachers are able to implement, evaluate, and transform it for their own teaching practice.

Mancha [19] found a lack of teaching practicums at initial teacher training centres, pointing out that didactic knowledge and specific training in PE were disconnected from subsequent classroom application. For this author [19], teachers lacked the didactic tools to carry out quality teaching. Mancha [19] pointed out gaps that teachers found upon arrival at secondary schools that may be due to those schools or to initial training received at university. The gaps identified were associated with teaching practices, didactics, didactic knowledge of content, students themselves, their interests and motivations, facilities, and resources.

Other papers in the Brazilian context such as De Souza Júnior, do Amaral, de Melo, Darido, and Lima [20], have pointed out difficulties in the production of PE teaching materials and have proposed improvement alternatives focusing on the design and use of materials. For De Souza Júnior et al. [20] some teachers in the field of PE are distrustful and unwilling to participate in the development of curricular materials to aid the teaching of certain sports included in the subject of PE. Despite their undeniable importance and contribution to the teaching-learning process, textbooks were highly criticized for supposedly transmitting a particular kind of ideology through their content [20]. These authors [20] explained that the historical development of PE in schools has little connection with textbooks, as the subject has been considered to focus on body movement reproduction and execution, rather than be a real curricular component. Nevertheless, PE textbooks are necessary in order to establish a critical relationship with PE and could help promote alternative teaching-learning models and curricular content among students. The results of the other authors [21] demonstrated that the examined PE texts showed a great divergence in terms of creativity, being necessary to apply policies that favor said creativity among the students.


*(b) Studies on the implicit ideological discourse in the images of PE textbooks*


This section involves research analysing the implicit content of images present in PE materials and textbooks. Absences and stereotypes have been identified as well as the limited and biased representation of certain groups. Guimarães Botelho and Garcia Neira [22] analysed textbooks in PE between 2000 and 2012 and highlighted the existence of stereotypes and prejudices in books by Brazilian and Spanish authors. Thus, several Spanish and Brazilian studies on body type depicted in textbook images concluded that stereotypes, prejudices, and discrimination prevailed with respect to gender, age, ethnicity, race, and body patterns.

Regarding the analysis of photographs in PE textbooks for secondary school, Táboas-Pais and Rey-Cao [23] and González and Rey [24] identified a clear inequity in the presence of male versus female images and in the physical activities attributed to each gender. Moya-Mata, Ruíz Sanchís, and Ros Ros [25] pointed out that the most frequent images on the covers of PE textbooks in terms of gender were mixed male-female (56%), followed by male (22%) and then female (12%). The male and female imbalance involving the male model and a gender-differentiated body culture was also pointed out by González-Palomares, Táboas-Pais, and Rey-Cao [26] and Táboas and Rey [23]. Their findings showed that female body culture was characterised by the practice of individual sports, artistic activities, fitness, and physical condition activities, in informal and utilitarian environments, indoor spaces, and at far from high levels of performance. On the other hand, male body culture was characterised by the practice of sports—both collective and individual—in competitive environments, outdoor spaces, and at elite levels of performance. The paper by Táboas and Rey [27] analysed the images of PE textbooks published in Spain between 2000 and 2006 and how they reflected gender differences. In the same vein, the findings by Abdelhay and Benhaddouche [28] showed that illustrations of men outnumbered those of women. In addition, following their traditional role in society, women were more often represented as teachers, secretaries, or top models, that is, in stereotyped activities. The men were shown as doctors, reporters, veterinarians, or actors, that is, in more outdoor, adventurous, and specialized activities, to which is added a dominion in PE and sports games. The findings revealed a need for increased awareness regarding textbook image content as well as overcoming gender stereotypes involving PE and sport. The results of other authors [29] showed a low presence of Olympic athletes in the textbooks, with more sexism in the text than in the images. These female athletes are mostly representative without disabilities, competing in individual sports modalities, with athletics being the sport preeminent.

A number of studies were found that examined the representation of race in PE textbooks for secondary school. Some of these involved case samples from different publishers and others involved textbooks under different educational legislation. Moya-Mata, Ruíz Sanchís, and Ros Ros [25] analysed 30 front pages of textbooks published under the Organic Law of Education (LOE) in Spain and highlighted current stereotypes regarding race and somatotype. Distribution by race showed that the highest percentage referred to the white race with 62%, compared to 38% for others [25]. The analysis of 2583 images by Táboas and Rey [23,30] showed that the representation of people whose look was akin to the in-group preponderated; that race played a role in the type of physical activity, the space, and the level of competition; and that, therefore, the analysed schoolbooks generated a stigmatized vision of racial diversity and their pictures reproduced and reinforced racial prejudices. This type of omission was also reported by studies analysing diversity in textbooks and didactic materials in general, where the habitual omission of certain subjects and social classes was highlighted [31]. In an analysis of 3855 images, González, Táboas and Rey [32] identified differences depending on which law was in force (e.g., races other than white are no longer depicted only in high-performance sports), but still found an under-representation of races other than white as well as an association of other races to sports such as basketball and athletics. Accordingly, the findings by Moya-Mata, Ros, Bastida, and Menescardi [33] revealed that PE textbooks perpetuated stereotypical models based on race and gender [33,34]. Hsu and Chepyator-Thomson [35] focused their study on the inclusion of multicultural education notions in curriculum and pedagogy in training for PE teachers of secondary school. The study focused on how high school PE textbook authors used multicultural education concepts. These authors [35] indicated that most of the textbooks treated multicultural education as an additional notion in the curriculum and highlighted gender and disability issues. In addition, the textbooks adopted multicultural educational approaches and raised in their content objectives gender, disability, and ethnicity issues, as well as equity-based pedagogy concepts.

In a similar line of research, the findings by González and Rey [24] highlighted how images perpetuate the invisibility of elderly and disabled people. Moya-Mata, Ruíz Sanchís, and Ros Ros [25] indicated that according to age distribution, 88% of images depicted children. Táboas, Rey, and Canales [36] demonstrated that elder people were under-represented in PE books and that both body characteristics and activities reflected a biased view of old age.

Other research analysed how pictures of impairment are represented in PE textbooks for high schools in Spain. An analysis involving 3316 images published in 36 textbooks by 10 publishers reported a remarkable disproportion between people with and without disabilities, and that, among the former, women were less represented. In addition, people with disabilities were represented as participants in a very restricted variety of competitive and elite sports activities [27,36]. In a more recent study [37] assessing 1094 images, many of the findings were validated: the “disabled” and “elderly” groups were under-represented, and there was a need to reduce the normalization of sedentary behaviours and minimize differences in gender, age, and disability categories in PE textbooks. After a content analysis of photographs in 59 PE textbooks, Hardin and Hardin [38] highlighted that general PE textbooks did not usually incorporate pictures of people with disabilities. González-Palomares, Rey-Cao, and Táboas-Pais [39] analysed how people with disabilities were reflected in PE textbooks, finding a practically invisible and biased depiction. Moya-Mata, Ruíz Sanchís, Ruíz, Alonso-Geta, and Ros [40], used content analysis to study the representation of disability in the images of PE textbooks. The analysis involved a sample of 3836 images that were gathered from textbooks published between 2006 and 2013 in Spain under the Organic Law of Education (LOE) [40,41]. The results revealed how far we are from the inclusion of people with disabilities. Likewise, Vidal-Albelda and Martínez-Bello [42] analysed how Spanish PE textbooks represented bodies with and without disabilities: only 10% of the pictures depict bodies with them, which were fully engaged in adapted sports activities. For these authors, alternative discourses on body diversity in schools and particularly in PE could be promoted if a critical analysis of curricular materials is carried out.

A study by Moya-Mata, Ros, and Royuela [43] analysing 997 images from nine textbooks published by five publishers found that male, white, slim, and juvenile bodies were emphasized. Interestingly, the authors reported that the scope and level of elitist physical activity had disappeared from textbooks, which represents a first step towards diversifying school physical activities. Guimarães Botelho and Garcia Neira [22] revealed that images of thin women and muscular men were predominant in textbooks from Brazil and Spain. By not taking into account ethnic, age, and multiform diversity, the pictures advocated white athletic people. González [44] reported similar results in terms of gender and Moya-Mata and Ros [45] pointed out a predominance of female figures who were young, white, and with an ectomorph somatotype in images from PE books. Another study [25] highlighted the predominance of the ectomorphic or thin somatotype, well above mesomorphic or endomorphic, regardless of the publisher analysed. Other research [46] showed the predominance of mixed groups of children, with ectomorphic bodies, of color or white race, and without disabilities, but, although gender diversity is represented, the materials continue to favor the reproduction of stereotypes and traditional hegemonic groups, perpetuating the invisibility of diversity in color or race, bodies and people with disabilities.

Finally, regarding activity type, Táboas-Pais and Rey-Cao [23] reported that sports get privileged treatment, while adapted physical activities practically did not appear. Likewise, in their analysis of textbook covers, Moya-Mata, Ros, and Chacón [47] encountered similar problems. Insofar as activity type, they reported sports to be the most widely represented (48%), followed by games (22%). Most images were linked to curricular content related to motor skills, games, and sports activities, while body expression, health, and knowledge of the body were less prevalent. Along these lines, González and Rey [48] pointed out that the majority of images referred to games and sports content and to physical activity models far from competition or high performance. Ismail, Soon and Kong [49] observed that illustrations and the center of mass concepts in textbooks were incorrect and could disorient the pedagogical practices of PE teachers and trainers. These authors suggested that in order to better reflect the true center of mass of gymnastic postures, future reference materials should incorporate anatomical and bio-mechanical knowledge in the preparation of figures.


*(c) Studies on the content of curricular materials from a historical perspective*


This research topic centres on the historical evolution of PE teaching materials. The studies in this area mainly focus on analysing the evolution of contents and activities included in textbooks and curricular resources for PE.

Galera [50,51] analysed research on school PE prior to 1939 observing that children’s games and educational gymnastics were recommended as the main contents, while walks and excursions were recommended to a lesser extent. PE books at the time emphasized activities that were typical of traditional PE, such as marches, races, and other outdoor activities. Towards the end of the study period, the inclusion of sports games as school content was pointed out. In the 1937 plan of the Republic, and never thereafter, alternating active guidelines with non-body matters were recommended [51]. In a similar line of research, Navarro [52] analysed changes in the way of understanding motor play and its application to school PE through a historiographic analysis of sociocultural and structural elements in 16 textbooks books for primary PE divided into two stages called traditional (1964–1982) and modern (1982 to 2002). The results point to changes across the two stages and to certain hegemonies in play structures with significant consequences for the pedagogical approach.

In this section, it is worth highlighting the research by Pastor [53], who carried out an extraordinary analysis of the role that manuals and textbooks played in the teaching of PE as well as the other aspects related to their characteristics and the conceptual definition of PE at each historical moment. Among the various conclusions, we would like to point out the following: *“There is no bibliographic inventory that identifies, catalogues, and controls the production of school textbooks and textbooks that have integrated the collection specifically dedicated to the teaching of School Physical Education”* [53] (p. 340).


*(d) Studies on health and the natural environment in PE teaching materials*


At present, the Sports Education model stands out among the teaching approaches that focus on students and have the greatest impact on PE professionals. The incorporation of basic competencies into the school curriculum has raised the need for defining the *health* competencies that students should attain by the end of compulsory education [54]. For Gómez [55], the main concerns involve health and lifestyles, seeing as sedentary behaviours have been associated with technological leisure. Learning possibilities and the didactic use of video games also play a part in this debate. Gilavand, Moosavi, Giolavand, and Moosavi [56] evaluated textbooks in Iran and noted that health education components received little or no attention.

From another perspective, the benefits of hybridization in sports education, with an approach based both on conventional and do-it-yourself materials, have not yet been explored either longitudinally or in primary education [57]. For these researchers, do-it-yourself materials encourage extracurricular physical activity and generate a high level of student enthusiasm. The authors reported [57] that students expressed a preference for do-it-yourself materials from the outset because of the ease of construction. Furthermore, after learning to make these materials at school, students could build them at home and thus practice in their free time. In addition, this study [57] reported more student physical activity during breaks when do-it-yourself materials were involved than with conventional materials. In any case, the use of both do-it-yourself and conventional materials in a hybrid way allows students to develop their motor skills and encourages physical activity at school and during free time. In another paper [58] involving a proposal for hybridizing the sports education model and the health-related PE model, the aim was to promote student autonomy (individual and group) as well as intra- and intergroup cooperation during the development of a didactic unit on skipping.

In research related to PE and the natural environment, Moya-Mata, Ruíz, Martín and Ros [59] analysed the sports activities in the natural environment depicted in 4339 images from 44 textbooks for primary school PE by eight Spanish publishers. A very scarce presence was found, with orienteering being the most often depicted (25.6%). Although depictions vary according to the publisher, it is necessary to include more and diverse representations of sporting activities in nature and promote increased social awareness [59].

García [60] proposed the incorporation of natural material resources into the educational system, both in the formal setting of PE classes and in informal leisure settings, as a good option for environmental education. This approach is purported to develop creativity, allow adaptation to different levels and be an excellent tool for cultural transmission. Also noteworthy is the study carried out by Baena and Calvo [61] presenting a practical experience in the construction of didactic materials for PE in the natural environment. Thus, we see that educational innovation in the area of PE should aim for culmination in activities in the natural environment. PE teachers try to adapt new teaching models in the natural environment to the specific contents and characteristics of daily work in the PE classroom, since the subject is situated outside the conventional curricular space. These authors [61] consider that the experience of preparing material for PE can reinforce all the educational and training potential of physical activity and foster proper student development.


*(e) Development of teaching materials by teachers themselves*


This section includes research analysing the characteristics of materials that have been prepared by teachers themselves for implementation in schools as a means of fostering student motivation during the teaching-learning process.

Huang, Chin, Hsin, Hung and Yu [62] developed an e-learning platform for PE, that offers sports-related courses including physical movements, rules, and first aid. The course uses digital multimedia materials including video, 2D animation, and 3D virtual reality. Through the use of this online learning platform, the user can gain knowledge about sports anywhere and anytime. According to these authors [62], the system is not only used to learn or improve sports ability but also helps to establish knowledge, develop e-learning platforms within PE, and provide didactic material for improved efficiency in the learning process. Along these lines, 3D video-assisted instruction has frequently been applied to physical sports; however, it does not involve interactive practice nor does it simultaneously incorporate textbook learning and sports skill practice [63]. The results of other authors [64] showed that the use of electronic textbooks has a positive effect on physical fitness.

Méndez [4] found the research on the self-made materials approach for PE to show positive psychological effects. Both the process and the product were very successful, improved learning, fostered creativity and personal participation, developed teaching skills, encouraged participation, inclusion, and cooperative learning, and promoted an education in values and the environment among PE students.

Zandoná and Silva [65] analysed the contents of body culture and its didactic use in the materials of PE subjects. They noted that the utilization of these resources was of crucial importance for giving greater meaning to teachers’ pedagogical work and favouring students’ understanding of physical exercise. This contributes to the strengthening and legitimization of PE as a pertinent curricular factor for the human, critical, conscious, and emancipated education of students.

Yet other studies have focused on projects for developing didactic materials for online training. In this regard, Rodríguez, Eirín and Alonso [66] highlighted the potential of multimedia and connected space, which make it possible to exchange experiences and perspectives, enriching materials and training multilingually. Some papers have proposed evaluation models to assess didactic materials prepared for PE and to establish criteria for the improvement of resources. Hongli, Junbo, Manli, and Zhihong [67], indicated that many shortcomings still exist in methods for evaluating PE teaching materials, making it difficult to fully reflect on the quality of PE textbooks and suggest improvements.

## 5. Conclusions

From the review of previous research we can draw the following conclusions:A good part of the research highlights the innovative potential of didactic materials for PE and how they are used. Studies have shown that didactic materials can contribute to facilitating learning as well as teacher planning and instructional work.Regarding the content of didactic materials, the reviewed research reveals a clear predominance of content involving motor skills and sporting activities as opposed to work related to body expression, knowledge, and health. Likewise, many papers show a clear inequity in the male versus female presence as well as differing activities for men and women. In general, images were found to perpetuate the invisibility of people with disabilities and the elderly.With respect to the treatment of diversity and educational inclusion, there does not generally seem to be a significant change in the treatment by publishers and there are clear cases of absence of racial and cultural diversity.The presence of the natural environment in educational resources continues to be scarce. Likewise, the existence of nature in textbooks and didactic materials for PE is treated unequally by the analysed publishers.With respect to body culture, the papers generally show a clear inequality in the presence of the male figure as opposed to the female figure in activity proposals and disability usually seems not to be taken into account.The need for teachers to develop their own materials in order to meet the diverse needs and interests of students should be emphasized. In general, textbooks can hardly meet the diversity of interests and needs and usually reflect a reality that has nothing to do with the context of students.With regard to the types of studies carried out and the analysis methodology applied, it should be noted that papers related to ideological discourse stand out, as do those carrying out a content analysis of textbooks and materials.

With a view to future research, it is recommended to continue research on teaching materials and PE books. This literature review attempted to minimize bias in the research protocol but it would be advisable for future research to conduct a systematic review meeting all the criteria of the PRISMA statement.

## Figures and Tables

**Figure 1 ijerph-19-07206-f001:**
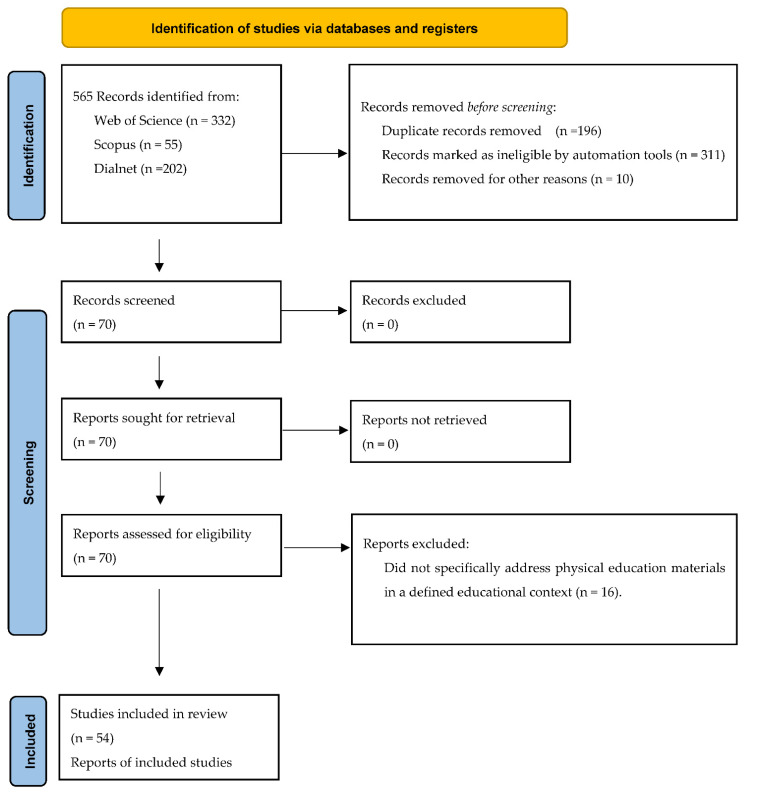
Flow diagram of the systematic search process.

**Table 1 ijerph-19-07206-t001:** Inclusion and exclusion criteria were used in the selection of publications.

Inclusion Criteria	Exclusion Criteria
1.a. Scientific documents published in original article format, reviews, doctoral theses, and books/ebooks	2.a. Scientific documents that do not have access to at least the abstract
1.b. Research that addresses PE materials	2.b. Scientific documents that address the materials but PE has not been contextualized
1.c. Research that addresses the materials in any of these educational stages: Early Childhood, Primary or Secondary Education	2.c. Scientific documents that address PE but do not provide details of the materials in any educational context
1.d. Scientific documents published until 31 December 2021 that are in English or Spanish at least in their title, abstract, and keywords	2.d. Duplicate items

**Table 2 ijerph-19-07206-t002:** Summary of the studies found on textbooks and learning materials in PE in the International context.

Categories	Number of Articles	Authors and Year
Studies on the use and characteristics of didactic materials and textbooks in PE	7	Barroso and Darido, 2016; De Souza Júnior, do Amaral, de Melo, Darido and Lima, 2015; Mancha, 2012; Perazas, Gil, Pardo and Soler, 2017; Peiró-Velert, Molina-Alventosa, Kirk and Devís-Devís, 2015; Ruíz, 2012; You, Lee, Craig, 2019
Studies on the implicit ideological discourse in the images of PE textbooks	28	Abdelhay and Benhaddouche, 2015; Botelho and Neira, 2014; González, 2005; González and Rey, 2013, 2015, 2019; González, Táboas and Rey, 2010; González-Palomares, Rey-Cao and Táboas-Pais, 2015; González-Palomares, Táboas-Pais and Rey-Cao, 2017; Guimarães and García, 2014; Hardin and Hardin, 2004; Hsu and Chepyator-Thomson, 2010; Ismail, Soon and Kong, 2018; Loro, Moya-Mata, Valencia-Peris, Nunes and Devís-Devís, 2021; Martínez-Bello and Molina-García, 2016; Moya-Mata and Ros, 2015; Moya-Mata, Ros, Bastida and Menescardi, 2013; Moya-Mata, Ros and Chacón, 2018; Moya-Mata, Ros and Royuela, 2016; Moya-Mata, Ruíz Sanchís and Ros Ros, 2017; Moya-Mata, Ruíz Sanchís, Martín Ruíz, Ros Ros, 2019; Moya-Mata, Ruíz Sanchís, Ruíz, Alonso-Geta and Ros, 2017; Ruiz Rabadán and Moya-Mata, 2020; Táboas-Pais and Rey-Cao, 2011; Táboas and Rey, 2012; Táboas and Rey, 2015; Táboas, Rey and Canales, 2013; Vidal-Albelda and Martínez-Bello, 2017
Studies on the content of curricular materials from a historical perspective	4	Galera, 2016, 2017; Navarro, 2006; Pastor, 2005
Studies on health and the natural environment in PE teaching materials	8	Baena and Calvo, 2008; Evangelio, Hurtado and Peiró, 2017; García, 2005; Gilavand, Moosavi, Gilavand and Moosavi, 2016; Gómez, 2016; Méndez, Martínez and Valverde, 2016; Moya-Mata, Ruíz, Martín and Ros, 2017; Torres, Marrero, Navarro and Gavidia, 2018;
Development of teaching materials by teachers themselves	7	Chang, Zhang, Huang, Liu and Sung, 2020; Hongli, Junbo, Manli and Zhihong, 2017; Huang, Chin, Hsin, Hung and Yu, 2011; Kubiyeva, Akhmetova, Islamova, Mambetov, Aralbayev and Sholpankulova, 2020; Méndez, 2018; Rodríguez, Eirín and Alonso, 2017; Zandoná and Silva, 2018

## Data Availability

Not applicable.

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
