# Peer review of "Textbooks and Learning Materials in Physical Education in the International Context: Literature Review"

_ijerph, 2022, doi:10.3390/ijerph19127206_

Round 1

Reviewer 1 Report

Dear Authors,

Thank you for taking up a very interesting topic of literature review on research related to textbooks and learning materials in physical education in the international context.  
In general, the study was designed and carried out correctly, although it requires some changes.

  • The abstract should precisely indicate the purpose of the study, why was it conducted? (similarly to lines 96-100).
  • In the abstract, the discussion is presented together with Conclusions /lines27/, which is not right, because they are separate parts of the manuscript.
  • In the Introduction /lines 42-51/, such a long quote is not necessary, especially with a reference to another source (Hernandez, 1990), which is not included in the references.
  • In the Material and methods, it is worth introducing more transparency. Perhaps the authors will consider introducing some theoretical background as a foundation for the analysis (e.g., Critical Interpretive Synthesis (CIS) or another approach).
  • In the same section, I also propose to introduce subsections such as: "search criteria" and "search procedure", as a result, the paper will be more readable.
  • In figure 1. The authors state that they removed 10 records for ‘other reasons’. It is worth to name these reasons in the text.
  • The Discussion section is not actually a discussion of the results, but a presentation of them. I propose a detailed presentation of the results to be moved to its place, i.e., in the Results section. In the Discussion, you should discuss the results in the context of the work of other authors, including other literature reviews. It is also worth mentioning the implications, both practical and theoretical.

I encourage you to revise this interesting manuscript.

Best regards,
The reviewer.

Author Response

After attending to all the comments made by the reviewers, we believe that the work has substantially improved in quality.

Here are the changes made based on your feedback:

Reviewer comments: Thank you for taking up a very interesting topic of literature review on research related to textbooks and learning materials in physical education in the international context.  
Authors' response: thank you for your comment.

Reviewer comments:  The abstract should precisely indicate the purpose of the study, why was it conducted? (similarly to lines 96-100).

Authors' response: Thank you for your comment. Added review objectives in abstract.

Reviewer comments:  In the abstract, the discussion is presented together with Conclusions /lines27/, which is not right, because they are separate parts of the manuscript.

Authors' response: It has changed. Now both parties have parted ways.

Reviewer comments:  In the Introduction /lines 42-51/, such a long quote is not necessary, especially with a reference to another source (Hernandez, 1990), which is not included in the references.

Authors' response: The selected text has been modified and is no longer a direct quote.

Reviewer comments:  In the Material and methods, it is worth introducing more transparency. Perhaps the authors will consider introducing some theoretical background as a foundation for the analysis (e.g., Critical Interpretive Synthesis (CIS) or another approach).

Authors' response: The figures and tables added in the text help to better understand the search and exclusion criteria applied in this review. Unfortunately, most of the works related to this topic were written in Spanish and no international reviews have been found. In order to have greater transparency in the manuscript, an attempt has been made to meet the maximum number of criteria recommended by the PRISMA protocol, following this article cited as [14] Page, M.J., McKenzie, J.E., Bossuyt, P.M., Boutron, I., Hoffmann, T.C., Mulrow, C.D. et al. The PRISMA 2020 statement: an updated guideline for reporting systematic reviews. BMJ 2021, 372:n71. https://doi.org/10.1136/bmj.n71

Reviewer comments:  In the same section, I also propose to introduce subsections such as: "search criteria" and "search procedure", as a result, the paper will be more readable.

Authors' response: Table 1 shows the inclusion and exclusion criteria of the different manuscripts. We believe that it is the clearest way to see these criteria.

Reviewer comments:  In figure 1. The authors state that they removed 10 records for ‘other reasons’. It is worth to name these reasons in the text.

Authors' response: The main argument of that exclusion has been added in the text.

Reviewer comments:  In the Discussion, you should discuss the results in the context of the work of other authors, including other literature reviews. It is also worth mentioning the implications, both practical and theoretical.

Authors' response: Modifications have been made in the discussion for a better understanding. Being a literature review and not having original results, the discussion was organized by categories in which the results and conclusions of the different studies analyzed were discussed.

Author Response

After attending to all the comments made by the reviewers, we believe that the work has substantially improved in quality.

Here are the changes made based on your feedback:

After attending to all the comments made by the reviewers, we believe that the work has substantially improved in quality.

Here are the changes made based on your feedback. Authors' response:

Thank you very much for your improvement comments included in the PDF document itself. The comments have been reviewed and have been modified in the attached text itself.

Reviewer 3 Report

The authors are to be acknowledged of the extensive literature review performed on the textbooks and learning materials mapped into physical education. Potential future research literature review might also shed light on learning/serious games' instruction on physical education across diverse educational levels.

Author Response

After attending to all the comments made by the reviewers, we believe that the work has substantially improved in quality.

Here are the changes made based on your feedback:

Reviewer comments: The authors are to be acknowledged of the extensive literature review performed on the textbooks and learning materials mapped into physical education. Potential future research literature review might also shed light on learning/serious games' instruction on physical education across diverse educational levels.

Authors' response: Thank you for your improvement comments for future revisions. In this it was proposed not to include more points because the length of the manuscript is considerable.

Round 2

Reviewer 1 Report

Dear Authors,

Thank you for the changes made and for referencing my remarks. In my opinion, the manuscript has gained both scientific quality and readability.

Currently, only editing issues remain to be improved - Figure 1 is illegible (fuzzy letters in part of the identification).

Regards,
The reviever